

# iCRESTRIGRS: A coupled modeling system for cascading flood-landslide disaster forecasting

Ke Zhang[1,2,3], Xianwu Xue[2], Yang Hong[2], Jonathan J. Gourley[4], Ning Lu[5], Zhanming Wan[2], Zhen Hong[2], Rick Wooten[6]

[1]Cooperative Institute for Mescoscale Meteorological Studies, University of Oklahoma, Norman, OK 73072, USA
[2]Hydrometeorology and Remote Sensing (HyDROS) Laboratory, School of Civil Engineering and Environmental Science, and Advanced Radar Research Center, University of Oklahoma, Norman, OK 73072, USA
[3]State Key Laboratory of Hydrology-Water Resources and Hydraulic Engineering, Hohai University, Nanjiang, Jiangsu, 210098, China
[4]NOAA/National Severe Storms Laboratory, Norman, OK 73072, USA
[5]Department of Civil & Environmental Engineering, Colorado School of Mines, Golden, CO 80401, USA
[6]North Carolina Geological Survey, North Carolina Department of Environmental Quality, Swannanoa, NC
28778, USA

*Correspondence to*: Ke Zhang (kezhang@ou.edu)

**Abstract.** Severe storm-triggered floods and landslides are two major natural hazards in the U.S, causing property losses of $6 billion and approximately 110-160 fatalities per year nationwide. Moreover, floods and landslides often occur in a cascading manner, posing significant risk and leading to losses that are

20  significantly greater than the sum of the losses from the individual hazards. It is pertinent to couple hydrological and geotechnical modelling processes toward an integrated flood-landslide cascading disaster early warning system for improved disaster preparedness and hazard management. In this study, we developed the iCRESTRIGRS model, a coupled flash flood and landslide disaster early warning system, by integrating the Coupled Routing and Excess STorage (CREST) model with the physically based Transient

25  Rainfall Infiltration and Grid-Based Regional Slope-Stability (TRIGRS) landslide model. The iCRESTRIGRS system is evaluated in four river basins in western North Carolina that experienced a large number of floods, landslides and debris flows, triggered by heavy rainfall from Hurricane Ivan during September 16-18, 2004. The modelled hourly hydrographs at four USGS gauge stations show generally good agreement with the observations during the entire storm period. In terms of landslide prediction in this

30  case study, the coupled model has a global accuracy of 89.5% and a true positive rate of 50.6%. More importantly, it shows an improved predictive capability for landslides relative to the stand-alone TRIGRS model. This study highlights the important physical connection between rainfall, hydrological processes and slope stability, and provides a useful prototype system for operational forecasting of flood and landslide.

35  **Key words:** Flood, Landslide, Warning System, Hazards, and Infiltration





## 1. Introduction

Severe flooding and landslides are two major natural hazards in the U.S and world. Flooding causes property losses of $3.7 billion and approximately 110 fatalities per year nationwide (Ashley and Ashley, 2008), while landslides are responsible for 25–50 deaths and damage exceeding $2 billion annually (Spiker and Gori, 2003). Shallow landslides induced by heavy rainfall have posed significant threats to human lives and property worldwide (Hong et al., 2006;Kirschbaum et al., 2010). Moreover, heavy rainfall, floods and landslides often occur in a cascading manner, where a relatively low-consequence event like heavy rainfall could trigger a severe flood and/or landslide that poses significant risk to an affected community and may lead to losses that are significantly greater than the sum of the losses from the hazards taken individually. One example is the intense precipitation in the Colorado Front Range on September 11-12, 2013 that triggered flash floods and at least 1,138 debris flows, resulting in 8 fatalities and more than 20,000 buildings, 485 miles of roads and 50 bridges either damaged or destroyed (Coe et al., 2014). Another example of the devastating impacts of cascading multiple hazards: rain from the remnants of Hurricanes Frances and Ivan triggered 400 reported slope failures of various types in the Blue Ridge Mountains of North Carolina, and at least 33 debris flows and major floods in Macon County, causing 5 deaths, destroying 16 homes, and damaging infrastructure based on the work of Wooten et al. (2008) and the following geological surveys. Such events make it critical to provide the public with risk-informed forecasting and warning systems in which multi-hazard threats are assessed and quantified.

According to a global natural disaster synthesis report (Dilley et al., 2005), over 790 million people are exposed to more than one natural hazard, based on the past two decades of historical loss data. Concurrent or time-lagged cascading multi-hazards are worldwide phenomena. In spite of their cascading nature, forecasts and warnings and risk assessments for such events conventionally are oriented towards single-hazards, treating the cascading events as independent phenomena (Hsu et al., 2011;Wastl et al., 2011). One example is the severe storm system accompanied by a deadly tornado, heavy rain, and flash flooding that occurred in Oklahoma City (OKC) on 31 May 2013, in which more people were killed unexpectedly by the flash flooding than by the tornado, marking it the deadliest flooding event that ever occurred in OKC. This is largely due to the fact that the storm (accompanied by heavy precipitation and the tornado) and flash flood were forecasted by two separate warning systems (Uccellini et al., 2014); the public's attention was mostly drawn to the tornado warnings (not to the flash flooding threat) mainly because this storm occurred only ten days after the disastrous EF-5 tornado which devastated Moore, OK and resulted in 24 fatalities and $2 billion in property damage. Although several recent studies have investigated multi-hazards and multi-hazard risk assessment (Budimir et al., 2014;Gill and Malamud, 2014;May, 2007;Mignan et al., 2014), these multi-hazard studies are still in the early stages of conceptual development (Gill and Malamud, 2014;Kappes et al., 2010). Knowledge gaps and disciplinary barriers in the development of multi-hazard approaches remain formidable. It is essential to understand the cascading effects of multiple natural hazards





in an integrated way in order to accurately forecast their occurrence and assess their potential risks and societal impacts.

Hydrological models have been used for operational flood forecasting since the development of the first
watershed hydrological model in 1966 (Crawford and Linsley, 1966). Hydrological models have evolved from lumped-process models (Williams and Hann, 1978;Sugawara et al., 1984) to semi-distributed models (Beven and Kirkby, 1979;Zhao et al., 1980) and fully distributed models (Abbott et al., 1986;Wigmosta et al., 1996;Wang et al., 2011). Several regional to global real-time flood forecasting systems using hydrological models as major tools have been implemented, including the NASA-University of
Oklahoma's Ensemble Framework For Flash Flood Forecasting (EF5) (Clark III et al., 2016), Flooded Locations and Simulated Hydrographs Project (FLASH) (Gourley et al., 2014), the European Commission Global Flood Awareness System (Alfieri et al., 2013), and the NASA-University of Maryland Global Flood Monitoring System (Wu et al., 2014), among others. In the last decade, physics-based, rainfall-triggered landslide models (Baum et al., 2010;Godt et al., 2009;Dietrich et al., 1995;Iverson, 2000;Liao et al.,
2010;Lu and Godt, 2008;Raia et al., 2014) have been developed to simulate slope stability influenced by topography, geology, and hydrological processes. Some pioneering studies have been conducted to couple hydrological models with landslide or slope stability models to link the hydrological process with soil mechanics. For example, Simoni et al. (2008) combined a distributed hydrological model called GEOtop with a geotechnical model for probabilistic estimation of landslide occurrence. Lanni et al. (2012) utilized a
dynamic topographic hydrological model to describe the subsurface processes and linked it with a simple hillslope slope stability model for modeling the initiation of shallow landslides. However, studies on dynamically coupling hydrological processes predicted by distributed hydrological models with soil physics and mechanics determining slope stability are still in a very early stage (Camera et al., 2013;Bogaard and Greco, 2014) due to lack of knowledge of interactions between these processes and
differences in the spatiotemporal scales of the flood and landslide events.

In this study, we present a framework that couples an established distributed hydrological model–Coupled Routing and Excess STorage distributed hydrological model (CREST) (Wang et al., 2011)–with a well-known landslide model–Transient Rainfall Infiltration and Grid-Based Regional Slope-Stability (TRIGRS)
(Baum et al., 2010)–to realize systematic and dynamical simulation of hydrological processes and their effects on slope stability. This integrated, coupled system is designed to serve as a prototype model for potential operational use. The objectives of this study are (1) to develop a coupled flood-landslide forecasting model system that can be forced by satellite- or radar-based Quantitative Precipitation Estimation (QPE) systems or can be easily forced with numerical weather prediction models or other
weather models, and (2) to evaluate the performance of this coupled modeling system in forecasting streamflow and slope failures.





## 2. Methodology and Data

### 2.1. Methodology

In this study, we developed an integrated modelling system in which the CREST distributed hydrological model is coupled with the TRIGRS landslide forecasting model (Fig. 1); therefore, the system is called

"integrated CREST-TRIGRS" or iCRESTRIGRS. The CREST and TRIGRS models are briefly introduced in the following two sub-sections, while the integration method is described in detail in Section 2.1.3.

### 2.1.1 CREST model

The CREST model is a grid-based distributed hydrological model developed by the University of Oklahoma (http://hydro.ou.edu) and NASA SERVIR Project Team (www.servir.net). It partitions net

precipitation into surface runoff and infiltration using the variable infiltration capacity curve (VIC), a concept originating from the Xinanjiang Model (Zhao, 1992;Zhao et al., 1980) and later represented in the VIC Model (Liang et al., 1996;Liang et al., 1994). Multi-linear reservoirs are used to simulate cell-to-cell routing of surface and subsurface runoff separately. The CREST model uses a cell-to-cell routing scheme to route overland flow to downslope cells where it is further partitioned to infiltration and overland flow

moving downslope using the VIC based runoff generation scheme; in this way, interaction between surface and subsurface water flow processes is accounted for (Wang et al., 2011). The SCE-UA (shuffled complex evolution method developed at The University of Arizona) optimization scheme (Duan et al., 1992) is implemented to automatically calibrate the distributed model parameters.

The CREST model has been widely used for regional to global studies, including flood inundation mapping over ungauged basins (Khan et al., 2011), statistical and hydrological evaluation of multi-satellite precipitation products (Xue et al., 2013), and detection and prediction of extreme flood events (Zhang et al., 2015). It has also been implemented in several operational systems, such as the FLASH (Flooded Locations And Simulated Hydrographs) project (http://www.nssl.noaa.gov/projects/flash/) and a near real-time global

hydrological simulation and flood monitoring demonstration system (http://eos.ou.edu).

### 2.2.2 TRIGRS model

The TRIGRS V2.0.06b model computes transient pore-pressure changes and attendant changes in the factor of safety (FS) due to rainfall infiltration using a two-layer system that consists of an unsaturated zone above a saturated zone (Baum et al., 2010). This model links analytical solution for transient, unsaturated, vertical

infiltration above the water table (Srivastava and Yeh, 1991) to pressure-diffusion solutions for pressure changes below the water table (Iverson, 2000). The solutions are linked through a transient water table that rises as water accumulates at the base of the unsaturated zone. Pore pressures computed by the models are subsequently used in one-dimensional slope-stability computations to estimate the timing and locations of slope instability (Baum et al., 2010). The TRIGRS model assumes that water can infiltrate with a maximum



infiltration rate, i.e., the saturated hydraulic conductivity ($K_s$) at each grid cell. It also has a simple surface runoff routing scheme for water movement from cells that have excess surface water to adjacent downslope cells where it can either infiltrate or flow farther down slope. Detailed description of the TRIGRS model can be found in Baum et al. (2008) and Baum et al. (2010).

**2.2.3 Integrated Model System**

We integrated the CREST and TRIGRS model through one-way coupling. In this way, the CREST model computes all hydrologic storages and fluxes, including interception by vegetation, infiltration, runoff generation, water routing, and re-infiltration of excess surface runoff from upstream cells to downstream cells, and provides the initial conditions, e.g. soil wetness and depth of water table (Fig. 1). The TRIGRS

model is implemented to compute pore-pressure and slope stability correspondingly. The coupling is seamlessly executed in a distributed fashion at every time step and continuously computes runoff, infiltration, factor of safety, pore-pressure, and other water balance components at each grid cell. As shown in Fig. 1, this integrated system has an open interface that provides a utility to couple this integrated model with any other Numerical Weather Prediction (NWP) model, climate model, or radar/satellite based

Quantitative Precipitation Estimation (QPE) system that provides precipitation and other necessary weather data to form an operational real-time forecasting or nowcasting system.

As mentioned above, the original TRIGRS model has its own schemes to estimate infiltration and runoff routing. In TRIGRS, it is assumed that runoff occurs when the precipitation and runoff supplied to a cell

exceed its infiltrability. The infiltrability is set to the saturated hydraulic conductivity ($K_s$) based on previous studies (Iverson, 2000;Hillel, 1982). The infiltration ($I$) at each cell is computed as the sum of the precipitation ($P$) plus any runoff from upslope cells ($R_u$) with the limitation that infiltration cannot exceed $K_s$ (Baum et al., 2010):

$$I = \begin{cases} P + R_u, & P + R_u \leq K_s \\ K_s, & P + R_u > K_s \end{cases}. \tag{1}$$

At each cell where $P + R_u$ exceeds $K_s$ the excess is considered runoff ($R_d$) and is diverted to adjacent downslope cells:

$$R_d = \begin{cases} P + R_u - K_s, & P + R_u - K_s > 0 \\ 0, & otherwise \end{cases}. \tag{2}$$

Overland flow between adjacent cells is assumed to occur instantaneously; thus, the rate of overland flow is not considered or computed in TRIGRS. TRIGRS enforces mass balance for each time step but does not

carry runoff over from one time step to the next or track water that enters storm drains (Baum et al., 2010;Baum et al., 2008).

Unlike TRIGRS, the CREST model is a distributed hydrological model, which specializes in modeling all major surface hydrological processes. The rainfall-runoff generation processes in CREST start from the

canopy interception. After $P$ passes the canopy layer, the excess precipitation that reaches the soil surface is





net precipitation ($P_{soil}$), which is further divided into surface runoff ($R$) and infiltration ($I$) according to the variable infiltration curve (VIC), a concept originating from the Xinanjiang Model (Zhao, 1992;Zhao et al., 1980) and later represented in the VIC Model (Liang et al., 1996;Liang et al., 1994). This model assumes that the point infiltration capacity $i$, which is the maximum water depth that can be stored in the soil column, varies over an area in the following relationship (Wang et al., 2011):

$$i = i_m \left[ 1 - (1 - a)^{\frac{1}{b}} \right], \tag{3}$$

where $i_m$ is the maximum infiltration capacity of a cell and is determined by soil properties; $a$ is the fraction of a grid cell and $b$ is the shape parameter. The amount of water available for infiltration can therefore be calculated as follows (Wang et al., 2011):

$$I = \begin{cases} W_m - W, & i + P_{soil} \geq i_m \\ (W_m - W) + W_m \cdot \left[ 1 - \frac{i + P_{soil}}{i_m} \right]^{1+b}, & i + P_{soil} < i_m \end{cases}, \tag{4}$$

where $W_m$ and $W$ are the cell's maximum water capacity and total mean water of the three soil layers, respectively. The overland and subsurface flows are further separated from excess rain governed by the saturated soil hydraulic conductivity. Then the cell-to-cell routing of the overland and subsurface runoff is simulated using the multi-linear reservoir method at each time step. CREST couples the runoff-generation process and the routing scheme to better represent the interaction between the surface and subsurface flow in three ways than simply routing surface flow downslope without considering its contribution to infiltration in downslope cells. First, overland runoff from upstream cells is treated as additional precipitation at the appropriate downstream cells available for infiltration and runoff partitioning. Second, soil water can be increased by lateral interflow from upstream cells. Third, channel runoff from upstream cells contributes to the receptor cell's overland reservoir depth. This implemented cell-by-cell runoff routing enables this model realistically compute the spatially and temporally varying values of runoff, soil moisture and infiltration. It also tracks the water movement through the basin.

In theory, replacing the simple infiltration and runoff routing schemes in TRIGRS with the more sophisticated runoff generation and routing methods in CREST will produce more realistic estimates of infiltration history in areas with nontrivial contribution of overland flow to infiltration. Moreover, the CREST model is able to simulate and provide necessary initial conditions and other inputs for the TRIGRS model. The iCRESTRIGRS model is, therefore, able to continuously, seamlessly simulate hydrological processes and solve pore-pressure and factor of safety for each cell at each time step with available forcing data. The minimum forcing data required to run this model are only precipitation and evapotranspiration, making it an easily implementable model.





### 2.2. Case Study and Data

#### 2.2.1 Study area and Case Study

We chose four adjacent river basins, the Upper Little Tennessee River basin, the Tuckasegee River basin, the Pigeon River basin, and the French Broad River basin, located in western North Carolina (Fig. 2a) as

our study area. The drainage area of the four basins ranges between 1,390 km$^2$ and 4,050 km$^2$.

Hurricane Ivan, the 10$^{th}$ most intense Atlantic hurricane ever recorded, passed through this region between Sept. 16 and Sept. 19, 2004. It triggered over 110 landslides across the study region, and at least 33 debris flows occurred in Macon County, causing 5 deaths, destroying 16 homes, and damaging infrastructure

(Wooten et al., 2008). Hurricane Ivan produced rainfall rates of 150-230 mm/h and precipitation totals from 52 to 351 mm across the study area (National Oceanic and Atmospheric Administration, 2004). The flood and landslide events triggered by the precipitation from Hurricane Ivan across western North Carolina serves as an ideal case study to test the integrated flood-landslide forecast system.

#### 2.2.2 Model Input Data

Data used in this study include radar-measured rainfall and satellite based estimates of actual evapotranspiration, digital elevation model (DEM), land cover and soil texture maps, observed river streamflow from gauges, and an inventory of landslide events (Bauer et al., 2012). All gridded data were either downscaled or aggregated to a spatial resolution of 3 arc-seconds (3″, i.e., ~90m) to ensure the forcing and auxiliary data match with each other. Bilinear interpolation is the method for spatial

downscaling in this study, whereas area-weighted resampling is used for aggregation.

The precipitation data were from the hourly, 4-km National Stage IV Quantitative Precipitation Estimation (QPE) product based on gauge and radar observations at NCEP (Lin and Mitchell, 2005). The Stage IV data were downscaled to 3″ using bilinear interpolation. The actual evapotranspiration (ET) data were

25 derived from a daily satellite remote sensing based ET record and are available at a spatial resolution of 8 km (Zhang et al., 2010;Zhang et al., 2009). The daily, 8 km ET data were first downscaled to a spatial resolution of 3″ using bilinear interpolation and further downscaled to hourly resolution using solar zenith angle as a function of solar declination, latitude, and hour angle of each grid cell.

The 3″ DEM (Fig 1.a), flow direction, and flow accumulation data were obtained from the USGS HydroSHEDS 3″ geo-referenced data sets (http://hydrosheds.cr.usgs.gov). These data serve as basic data for the distributed iCRESTRIGRS model to establish topological and geomorphological connections among grid cells and derive further topographical information such as slope (Fig. 1b). The map of soil type was from the State Soil Geographic (STATSGO), distributed by the Natural Resources Conservation

Service (NRCS) of the US Department of Agriculture. The soil texture classes were converted to 12 USDA





soil texture classes plus rock and organic matter. The land cover map was derived from the 30-m National Land Cover Database (NLDC) 2011 land cover database (Homer et al., 2015).

15-minute streamflow observations from four USGS streamflow gauges (#03503000 at Little Tennessee River, # 03513000 at Tuckasegee River, # 03460795 at Pigeon River, and # 03453500 at French Broad River) were aggregated to hourly resolution and serve as streamflow validation data for the model. The locations of landslide events were identified by the North Carolina Geological Survey through field surveys and other remote-sensing techniques (Wooten et al., 2008;Bauer et al., 2012).

### 2.2.3 Model Parameters and Initialization

As the developed prototype system is aimed for operational use over a large region, we determined the values of model parameters at each grid cell based on its soil class rather than using very limited local measured values. For this purpose, we built a parameter look-up table based on the USDA textural soil classification. Table 1 summarizes the values of key common parameters used in both TRIGRS and iCRESTRIGRS. The values of all these parameters were roughly estimated as the means of their value ranges determined from the literature.

Because all model parameters were determined from soil types *a priori*, we did not conduct model calibration in this study. To minimize the uncertainty in the initial conditions, the iCRESTRIGRS model was spun-up for one year beforehand.

### 2.3 Statistical Metrics for Evaluating Model Performance

To evaluate the performance of this integrated model system, we applied a suite of statistical metrics to evaluate the model results. We computed relative bias, Pearson correlation coefficient (CC), and Nash–Sutcliffe Coefficient of Efficiency (NSCE) for the modeled hourly discharge series using the four USGS gauge stations.

To quantitatively measure the predictive capabilities of the models for landslide prediction, we calculated the confusion matrix through comparison between the binary predictions of slope failure and the landslide inventory database. The confusion matrix consists of four possible outcomes (Fawcett, 2006): (1) a modeled landslide is a true one (True Positive, TP), (2) a modeled landslide is a false one (False Positive, FP), (3) an observed landslide is not captured by the model (False Negative, FN), and (4) a grid cell is stable in both model and observation (True Negative, TN). Based on the confusion matrix, a series of indices can be calculated:

$$TPR = TP/(TP + FN), \tag{5}$$

$$TNR = TN/(TN + FP), \tag{6}$$



and  $Efficiency = (TP + TN)/(TP + TN + FP + FN),$ (7)

where TPR is the True Positive Rate and also called sensitivity, and TNR is the True Negative Rate and also called specificity. The sensitivity statistic measures the percentage of positive cases correctly predicted, while the specificity statistic quantifies the percentage of negative cases correctly predicted (Begueria, 2006;Fawcett, 2006). The Receiver Operating Characteristic (ROC) curve analysis was further applied to evaluate model results for landslide predictions and to compare differences between the TRIGRS and iCRESTRIGRS models. A ROC curve consists of TPR and TNR pairs, which are computed from the respective confusion matrices for different cutoff values. In our case, the cutoff variable is FS. A ROC curve shifted towards the upper-right corner means better model performance. The better the performance of the model the larger is the Area Under the ROC Curve (AUC); therefore, the AUC index serves as a global statistical accuracy for the model.

## 3. Results

### 3.1 Characteristics of Hurricane Ivan Induced Storm

Hurricane Ivan passed through western North Carolina between Sept. 16 and 18, 2004. The storm in this region started around 11 UTC September 16, 2004 and completely ceased around 3 UTC September 18, while the majority of rainfall occurred in the first 24 hours  (Fig. 3a). It brought an average rainfall of ~130 mm within 24 hours across the region (Fig. 3a), while accumulated rainfall reached maximum values in the southern parts of the four river basins (Fig. 3b). The storm roughly moved from southwest to northeast (Fig. 3c); differences in peak time of rainfall across this region can be as large as five hours (Fig. 3c). Apparently, the storm was rapid and intense.

### 3.2 Model Evaluation and Comparison

The modeled hourly discharge series between September 16 and 24, 2004 by the iCRESTRIGRS model were compared with the observations at the four USGS gauge stations (Fig. 4a-d). The modeled hydrographs show generally good agreement with the observations. CC is larger than 0.80 at all stations. Relative bias falls within ±34%. The NSCE values at three stations except the one located at French Broad River are larger than or equal to 0.65. The low NSCE value at the French Broad River is largely due to a time shift between modeled and observed peak discharges (Fig. 4d). In general, the above results indicate that the iCRESTRIGRS model is capable of simulating runoff process well and predicting the flood events. In the model, whenever the FS value is less than 1.0, the land surface's slope is predicted to fail and there is a corresponding landslide. Fig. 5a and Fig. 5b show the maps of minimum FS values during the whole storm period modeled by TRIGRS and iCRESTRIGRS, respectively. In Fig. 5, the reported landslide events are plotted as magenta circles. The reported landslide events in the inventory are just point data, but the actual landslides usually occur over areas ranging in size. Therefore, we regard that the model successfully predicts a real landslide if one or more cells within a radius of 500 m around a reported



landslide point have a FS value of < 1. The spatial distribution of reported landslides generally corresponds with the spatial patterns of model minimum FS values (Fig. 5a,b). In other words, the actual landslides are mostly located in the areas in which models predict unstable or close to unstable conditions (Fig. 5a,b). A notable difference in the spatial patterns of FS by the two models is that more areas in TRIGRS (Fig. 5a)

have unstable slopes than in iCRESTRIGRS (Fig. 5b). This confirms that factor of safety computed by the models is sensitive to hydrological processes, in particular the infiltration process, because the largest difference between TRIGRS and iCRESTRIGRS lies in the way how infiltration and runoff routing are computed. For the TRIGRS model, the TPR, TNR, and accuracy statistics are 50.1%, 83.4%, and 83.2%, respectively, when FS=1 is set as a cutoff value for slope stability. For the iCRESTRIGRS model, the three

metrics are 50.6%, 89.6%, and 89.5%. These results indicate that the iCRESTRIGRS model shows better results and that coupling the CREST distributed hydrological model with the TRIGRS model leads to an improved model performance at least for this case study.

The ROC analysis demonstrates that the coupled system generally has higher sensitivity and specificity

relative to the original TRIGRS model (Fig. 6). The AUC values for the TRIGRS and iCRESTRIGRS models are 0.75 and 0.79, respectively, suggesting that iCRESTRIGRS is a better model than TRIGRS. As mentioned above and shown in Fig. 7, the largest difference between the two models lies in the model infiltration values. It is clear that the simple infiltration and rain excess routing schemes implemented in the TRIGRS model leads to higher values of infiltration than in the iCRESTRIGRS model (Fig. 7a). The

regional average accumulated infiltration during the Sept. 16-18 storm modeled by TRIGRS is 104.9 mm, while the value modeled by iCRESTRIGRS is just half of it, i.e. 52.2 mm (Fig. 7a). The larger infiltration rate in TRIGRS than in iCRESTRIGRS appears across the whole region (Fig. 7b,c). This explains why the TRIGRS results have generally lower FS values than the iCRESTRIGRS results and why TRIGRS has a higher false positive rate than iCRESTRIGRS (Fig. 5).

**3.3 Evolution of Modeled Cascading Flood-Landslide Hazards**

We further investigated the evolution of the storm in terms of accumulated rainfall during a 6-hour period and the corresponding responses of hydrological processes (e.g., infiltration and overland runoff), slope stability, and pressure modeled by the iCRESTRIGRS model (Fig. 8). In Fig. 8, the rainfall and infiltration are 6-hour accumulated values, while overland runoff is the average value during each 6-hour period.

Factor of safety is the minimum value during each period, while pore-pressure is the value at the depth and at the time corresponding to the lowest FS.

During the first 6-hour period, rainfall intensities are low across the region. The accumulated rainfall is generally less than 10 mm during this period (Fig. 8). In response to this, modeled infiltration rate and its

accumulated value are low as well, while simulated overland runoff mainly appears in the main river channels. Very few unstable slopes appear across this region in the model and pore-pressure is generally


low. During the second 6-hour period, rainfall rate and modeled infiltration rate increase to some extent, especially in the Little Tennessee River and Tuckasegee River basins. The model results show that overland flow starts to appear in the small tributaries and creeks (Fig. 8). Pore-pressure reaches high values in some areas of the Little Tennessee River and Tuckasegee River basins, resulting in some unstable slopes.

These unstable slopes are largely located in the areas with steep slopes. During the third 6-hour period, rainfall reaches maximum and has an accumulated value larger than 30 mm over most of the region. Correspondingly, infiltration rates also reach maximum and overland runoff appears everywhere with rapid rises of stream flow in the drainage network. The number of modeled landslides has increased dramatically accompanied by large increases in pore-pressures. As the storm enters the fourth 6-hour period, rainfall and

infiltration intensities decline but still maintain high levels. Pore-pressures in some regions continue to rise, resulting in some new landslides. During the fifth period, rainfall and infiltration intensities reduce greatly. Runoff on the land and in many upstream reaches of these rivers starts to subside. Pore-pressure declines in many areas but remain high in some areas in response to accumulative infiltration processes. The number of modeled landslides during this period also decreases. The detailed analyses of rainfall, and modeled

hydrological and geotechnical responses on a phase-by-phase basis show that the model results show reasonable responses to the evolution of the storm in space and time. It also emphasizes the cascading nature of rainfall-triggered floods and shallow landslides.

## 4. Conclusion and Discussion

This study presents a new, coupled model system, which integrates the CREST distributed hydrological
model with the TRIGRS landslide model for flood and landslide forecasting. Driven by the hydrological states and fluxes modeled by CREST, iCRESTRIGRS improves over TRIGRS by the providing more accurate initial conditions such as degree of soil saturation and depth of the water table. Furthermore, CREST specializes in the simulation of hydrological processes and fluxes and can thus provide more realistic hydrological fluxes such as infiltration for TRIGRS, leading to better accuracy for landslide
forecasting. The case study demonstrates that the integrated model shows better results than the stand-alone TRIGRS model for landslide forecasting.

The modelling system presented in this study is also developed as a framework and is able to adopt other hydrological models and landslide models as alternatives to compute hydrological processes and soil
stability. Therefore, this can be easily expanded to build an ensemble-based system. This coupled modelling system has low requirements for input data as well, making it easy to couple with other numerical weather prediction models and real-time QPE forcings.

It is worth to note that there is still a large room for improving the predictive capabilities of iCRESTRIGRS
for flood and landslide forecasting. In particular, the true positive rate for landslide forecasting in iCRESTRIGRS in the case study is not high. This can be improved through further parameter optimization





and/or implementation of reliable pedotransfer functions. In addition, slope stability is highly dependent on the slope. In this study, the grid spacing is set to 90-m rather than a finer resolution because we were limited by computational burden; plus, this prototype system is designed from operational use, so it is impractical to run this system with an extremely fine resolution over a large region. However, a nested

modeling approach, which executes the hydrological model at a coarser resolution, and allows the landslide model to be executed at finer and coarser resolutions in the landslide prone areas and stable areas, respectively. Additional evaluation of this model in larger regions and under different conditions will better support the predictive capability and robustness of this model.

**Acknowledgement**

This research was funded by NOAA/Office of Oceanic and Atmospheric Research under NOAA-University of Oklahoma Cooperative Agreement #NA14OAR4830100, and a grant (NNH10ZDA001N-ESI) from the NASA Surface and Interior program.

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

**Figure 1.** The framework of the proposed integrated CREST and TRIGRS modeling system, which is able to forecast flood and rainfall-triggered landslide events.





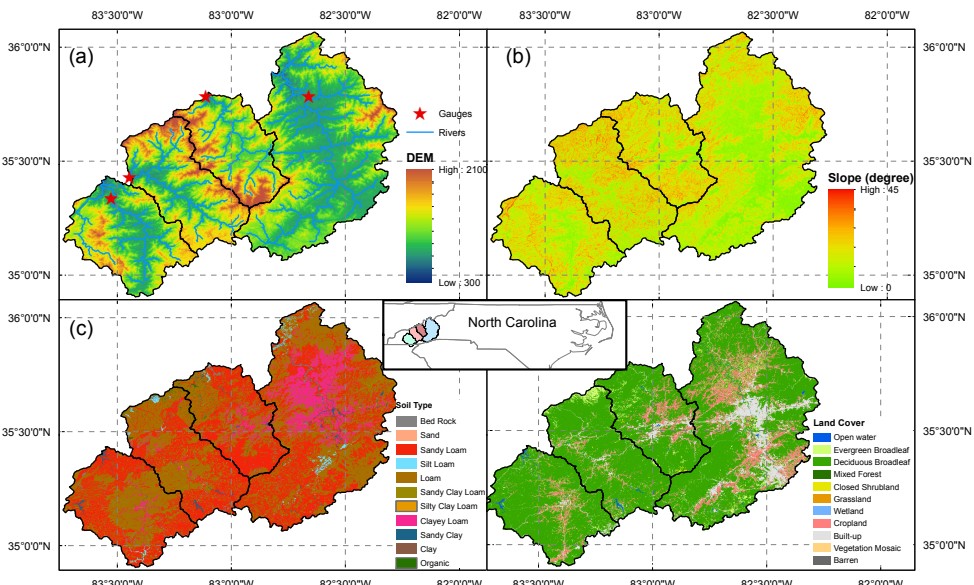

**Figure 2.** (a) Elevations, (b) slopes, (c) soil types, and (d) land cover types of the study region, which
include four river basins, the upper Little Tennessee River basin, Tuckasegee River basin, Pigeon River
basin, and French Broad River Basin; the inset shows the locations of the four basins within North Carolina.



**Figure 3.** (a) Accumulated and hourly regional-average rainfalls during the Sept. 16-18 storm period in the study region, (b) spatial pattern of accumulated rainfall during the storm period, and (c) spatial pattern of

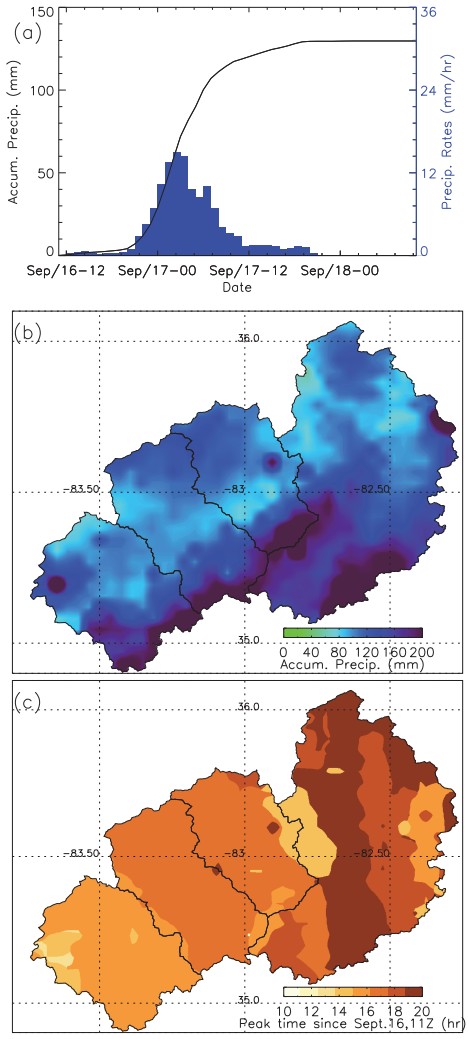

5    rain peak times since Sept. 16, 11 UTC.




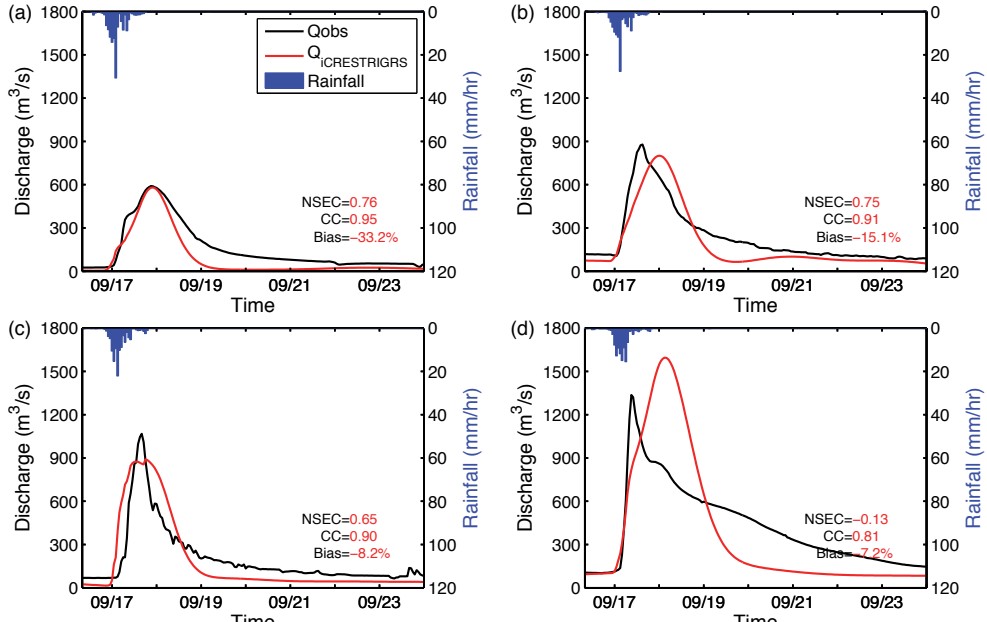

**Figure 4.** Observed and modeled hydrographs and basin-average rainfall rates at (a) the upper Little Tennessee River basin, (b) Tuckasegee River basin, (c) Pigeon River basin, and (d) French Broad River basin.




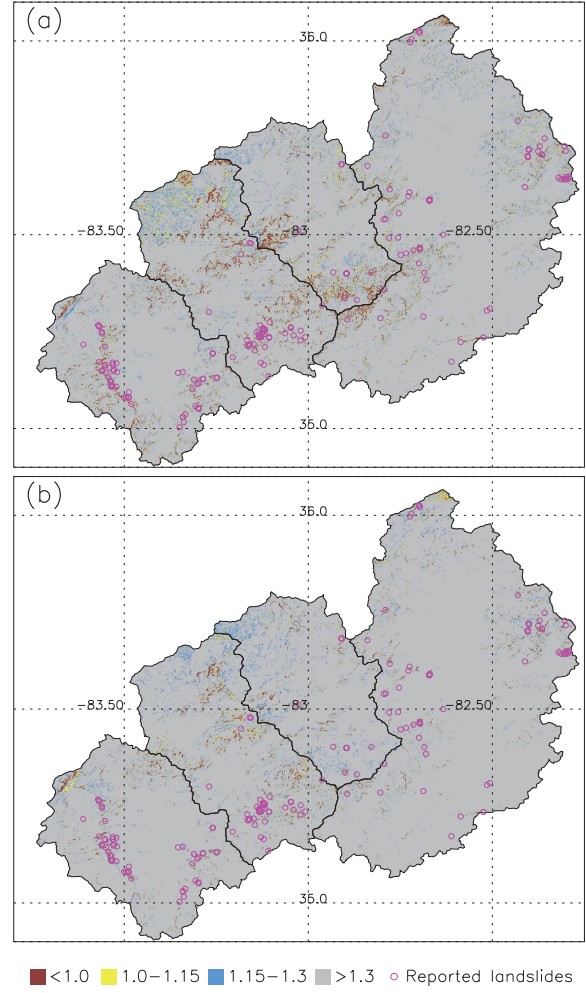

**Figure 5.** Comparisons (a) between TRIGRS modeled FS and reported landslide events, and (b) between iCRESTRIGRS modeled FS and reported landslide events.




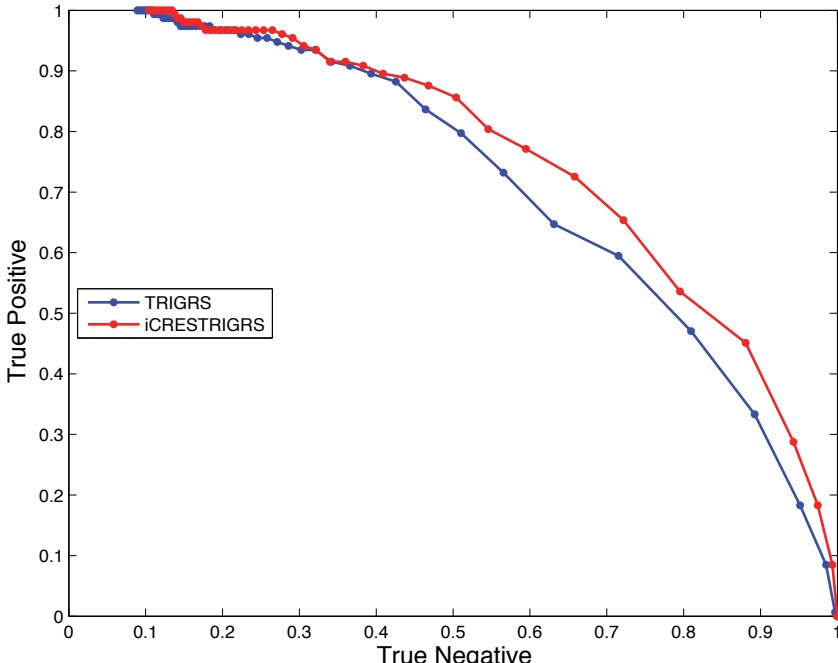

**Figure 6**. Receiver operator characteristic (ROC) graph comparing slope stability results from the TRIGRS and iCRESTRIGRS models.





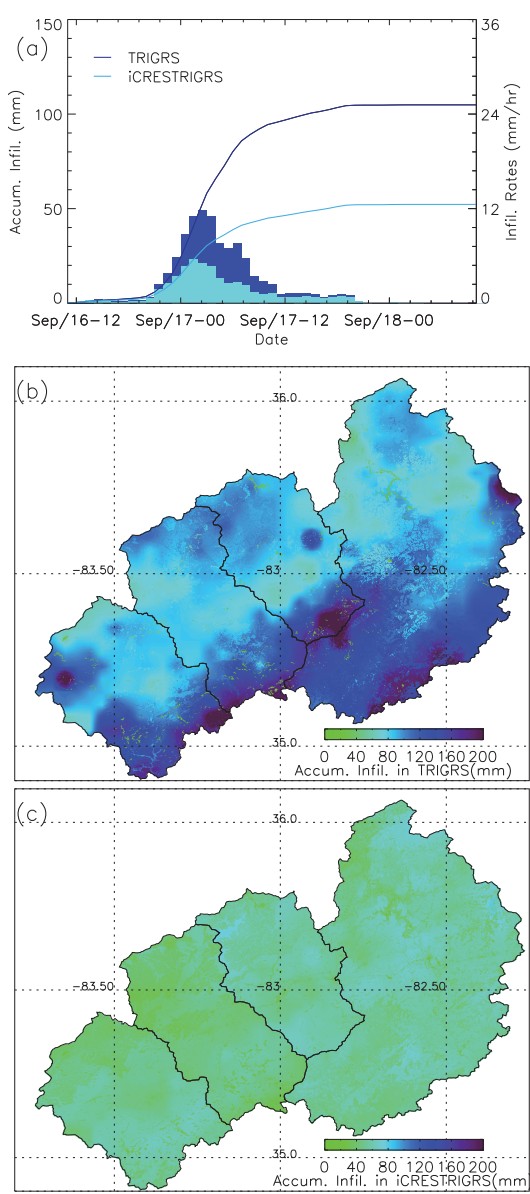

**Figure 7**. (a) Accumulated and hourly infiltration values from the TRIGRS and iCRESTRIGRS models, and spatial patterns of accumulated infiltrations from the (b) TRIGRS and (c) iCRESTRIGRS models.





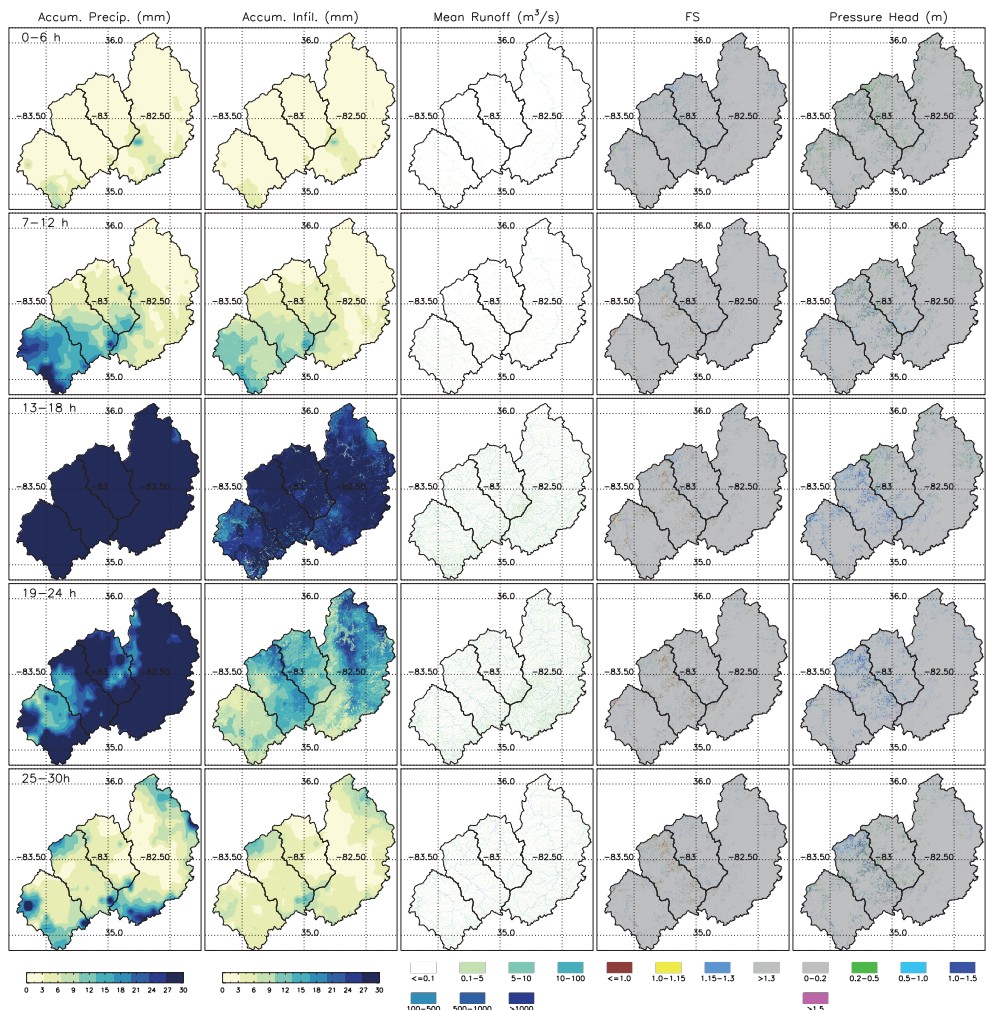

**Figure 8**. Maps showing the evolution of storm (rainfall), hydrological responses such as infiltration and runoff, slope stability, and pressure head at the depth corresponding to the lowest factor of safety.




**Table 1.** Summary of key parameter values used in both TRIGRS and iCRESTRIGRS in this study.

| USDA Soil Texture Type | Soil Cohesion[1] (kPa) | Porosity[1] | Saturated Hydraulic Conductivity[2] (m/s) | Friction Angle[1] (degree) | Soil Dry Unit Weight[1] (kN/m³) |
|---|---|---|---|---|---|
| Sand | 5.0 | 0.43 | $2.44\times10^{-5}$ | 40.0 | 21.0 |
| Loamy sand | 7.5 | 0.42 | $1.78\times10^{-5}$ | 28.5 | 20.5 |
| Sandy loam | 6.0 | 0.40 | $1.02\times10^{-5}$ | 32.0 | 15.0 |
| Silt loam | 9.0 | 0.46 | $2.50\times10^{-6}$ | 24.0 | 14.0 |
| Loam | 10.0 | 0.43 | $4.53\times10^{-6}$ | 22.5 | 13.0 |
| Sandy clay loam | 29.0 | 0.39 | $6.59\times10^{-6}$ | 20.0 | 15.0 |
| Silty clay loam | 50.0 | 0.48 | $1.44\times10^{-6}$ | 16.5 | 14.0 |
| Clayey loam | 35.0 | 0.46 | $2.72\times10^{-6}$ | 20.0 | 14.0 |
| Sandy clay | 24.5 | 0.41 | $4.31\times10^{-6}$ | 22.5 | 18.5 |
| Silty clay | 30.0 | 0.49 | $1.06\times10^{-6}$ | 18.5 | 18.0 |
| Clay | 40.0 | 0.47 | $1.31\times10^{-6}$ | 16.5 | 19.5 |
| Silt | 9.0 | 0.52 | $2.05\times10^{-6}$ | 26.5 | 16.5 |

[1] Values were roughly set to the means of the ranges determined from Das (2008), Hough (1969), Terzaghi et al. (1996) and (Dysli, 2000);

[2] Values were estimated by the pedotransfer equations of Cosby et al. (1984) using the mean sand and clay fractions of each soil class.