# Peer review of "iCRESTRIGRS: A coupled modeling system for cascading flood-landslide disaster forecasting"

_Hydrology and Earth System Sciences, 2016_

## Referee Comment (RC1) · Anonymous Referee #1 · 16 May 2016

General Comments: This paper represents a significant conceptual advancement in the modeling of flash flooding and shallow, rainfall-induced landslides. Combining CREST and TRIGRS appears to provide a fairly complete solution for modeling the surface water, shallow subsurface hydrology related to runoff and flooding as well as the initial hydrologic conditions, transient pressure head, and slope stability processes related to landslide initiation. The paper is generally well written and provides a general framework for modeling the hazards resulting from heavy or prolonged rainfall. The paper could be improved by stating more of the modeling assumptions (see specific comments) used in the case study and making a number of technical corrections.

Specific Comments:

[Figure]

1. P. 1, lines 19-20 and p. 2, line 9: The phrase "leading to losses that are significantly greater than the sum of the losses from the individual hazards" seems a little nonsensical. I understand that the authors are trying to convey the synergistic effects of flooding and landslides, but the effect of combined versus separate action of the hazards needs to be expressed more clearly. Consider "losses resulting from the combined hazards are significantly greater than the sum of losses from the hazards if acting separately"

2. P.1 line 23. It is confusing to refer to this combined modeling system as an "early warning system." A modeling system is part of many early warning systems, but warning systems consist of much more than computer models. Change "coupled flash flood and landslide disaster early warning system" to "coupled flash flood and landslide initiation modeling system"

3. P. 2, lines 27 – 32. These two sentences seem somewhat contradictory. The emotional effects of the recent devastating tornado seems to have been more critical in determining public behavior than the source of the warnings. Either rephrase to clarify that different sources of the flash flood and storm forecasts had a lesser role or explain how this contributed to the fact that the public's attention was drawn to the tornado warnings.

4. P. 4, line 12. Is there a reference for the "multi-linear reservoir" concept?

5. P. 4, lines 30 – 31. Delete reference to Iverson (2000) in line 31. The "pressure-diffusion solutions for pressure changes below the water table," though somewhat similar to the pressure-diffusion solution presented by Iverson (2000) for specified flux at the ground surface use different boundary conditions.

6. P. 6, line 6. Are the parameters a and b in equation 3 determined theoretically or empirically?

7. P. 8, lines 10 – 20. A few more details are needed regarding modeling assumptions: What assumptions were made regarding soil depth (constant or spatially varying? if

so, how?)? Did you check the factor of safety for prestorm initial conditions to confirm that no (or very few) grid cells had a factor of safety less than 1? For the pressure head computations used to compute factor of safety, did TRIGRS and iCRESTRIGRS use the unsaturated infiltration model as described at the bottom of p. 4? If so, why does table 1 not contain columns for the alpha (inverse height of capillary fringe) and residual moisture content values used in computing pressure head?

8. P. 12, lines 6 – 8. A new MPI version of TRIGRS is available that could help with the large area and finer grid assessments suggested here. The citation is Alvioli, M., and Baum, R.L., 2016, Parallelization of the TRIGRS model for rainfall-induced landslides using the message passing interface: Environmental Modelling & Software, Vol. 81, July, p. 122 - 135. doi:10.1016/j.envsoft.2016.04.002

9. Figure 6. Please label key factor of safety values (0.9, 1.0, 1.1) along the ROC curve.

Technical Corrections:

P. 2, lines 16 – 17. Change "damaging infrastructure based on the work of Wooten et al. (2008) and the following geological surveys." to "damaging infrastructure (Wooten et al. 2008; Bauer et al. 2012)."

P. 2, line 27. Change "marking" to "making"

P. 2, line 28. Change "largely" to "partly"

P. 2, line 29. Change "(Uccellini et al., 2014); the public's attention" to "(Uccellini et al., 2014). Moreover, the public's attention"

P. 6, line 21. Change "model realistically compute" to "model to realistically compute"

P. 7, line 34. Insert "dataset" after "(STATSGO)"

P. 8, line 2. Change "Land Cover Database (NLDC) 2011 land cover database (Homer et al., 2015)." to "Land Cover Database (NLDC) 2011 (Homer et al., 2015)."

P. 8, lines 4 – 6. Change "15-minute streamflow observations from four USGS stream-flow gauges (#03503000 at Little Tennessee River, # 03513000 at Tuckasegee River, # 03460795 at Pigeon River, and # 03453500 at French Broad River) were aggregated to hourly resolution and serve as streamflow validation data for the model." to "Streamflow observations from four USGS streamflow gauges (#03503000 at Little Tennessee River, # 03513000 at Tuckasegee River, # 03460795 at Pigeon River, and # 03453500 at French Broad River) were aggregated from 15-minute to hourly resolution and serve as streamflow validation data for the model."

P. 9, line 11. Insert "indicator" after "global statistical accuracy"

P. 9, lines 19 – 20. Change "Apparently, the storm was rapid and intense." to "The storm was rapid and intense."

P. 9, line 29. Change "the land surface's slope is predicted to fail" to "the regolith that covers the sloping ground surface is predicted to fail"

P. 10, line 7. Change "way how infiltration" to "way that infiltration"

P. 11, line 34. Change "It is worth to note that there is still a large room for improving" to "It is worth noting that there is still much room for improving"

P. 13, line 1. Change "and Anders, C. F.:" to "and Anderson, G. F.:"

Table 1. If the unsaturated infiltration option was used in TRIGRS and iCRESTRIGRS, then add columns for the inverse height of capillary rise (alpha) and residual moisture content (theta-sub-r).

---

## Referee Comment (RC2) · Anonymous Referee #2 · 19 May 2016

Dear Editor, dear Authors,

I read with interest this research paper which describes a coupled system for prediction of landslide occurrences triggered by intense precipitation. The system is based on the use, in a cascade manner, of the models CREST and TRIGRS. Finally, the authors describe the results of a interesting case study. In the Introduction, the issues of flood and landslide disasters are well described in the context of natural hazarda which characterize the study area.

The general objects of the work are clear and well stated, the proposed methodology is of good scientific interest, and also the presentation quality is satisfactory. However, I have some doubts on the efficiency of the created system and I argue the use of the

terms 'early warning system' and 'flood'.

Please, read in the following my GENERAL COMMENTS

FLOOD and EARLY WARNING

1. I am not sure if using the term 'flood' is here appropriate. If I understood correctly, CREST model predict the discharge at outlet and the spatially distributed surface runoff. I did not see here any flood disaster forecasting analysis (as claimed by the authors), which normally expects identification of floodplains, critical discharges, etc.. Does the model include a propagation module? Also, note that in flash flood events, the evapotraspiration is almost negligible, whereas the spatially distributed component of CREST model mainly derives, if I understood correctly, from the use of LAI and vegetation cover, from which evapotraspiration and interception processes are estimated.
2. Similarly, an early warning system is a more complex system which includes the definition of a 'chain of different communication systems working together and aimed at the detection, analysis and mitigation of potentially hazardous events'. For example, they normally includes a monitoring network for real-time analysis and the definition of thresholds for launching alert/waning signals. I would simply say that the developed system could be potentially used within an early warning system'(P1L23)

LITERATURE REVIEW 3. Authors state that 'studies on dynamically coupling hydrological processes predicted by distributed hydrological models with soil physics and mechanics determining slope stability are still in a very early stage' (see discussion at P3L16-25). Actually I do not agree with authors analysis. Literature provides many examples of spatial distributed and coupled hydrological-stability models which have been successfully utilized. Besides the ones cited by the authors (Simoni et al., 2008 and Lanni et al., 2012), authors can refer, for example, to Burton and Bathurts (1998), Claessens et al. (2007), Arnone et al. (2011), Lepore et al. (2013), Tao and Barros (2014) (these last two are also cited and described by Bogard and Greco (2014) as 'good examples' of coupled physically based hydrological and slope stability modeling at catchment scale). Also, you may more deeply discuss the sentence 'due to lack of knowledge of interactions between these processes and differences in the spatiotemporal scales of the flood and landslide events'. CREST MODEL 4. If I understood correctly, CREST model does use only precipitation, LAI and vegetation cover as distributed in input variables (as said above). Other model parameters I guess are homogenous across the watershed. How about the soil parameters that influence the infiltration processes, and slope stability, such as porosity, hydraulic conductivity, retention curve parameters,? Please specify whether these are distributed or homogeneous.

TRIGRS MODEL 5. There is no mention on how FS is computing and which are the parameters affecting stability. I would like to see the FS equation, explain whether is based on classical Mohr-Coulomb or Bishop failure criterion (for unsaturated conditions) and if it is computed ad a fixed soil thickness of different soil depths. Indeed, FS varies significantly with depths, at given parameters (see other comments below).

INTEGRATED SYSTEM 6. The integration between the two models is not very clear to me. TRIGRS model has its own infiltration module which is based on an analytical solution of the Richards equation for vertical infiltration. As input, it requires rainfall intensity; based on the hydraulic properties, which vary in space, it computes the infiltration rate and thus the pore-pressure (or soil moisture), which are used to compute the FS. My question is, how this framework interfaces with CREST model? My impression is that the two models are simply used in 'cascade' and therefore they are run separately, to evaluate first the runoff, and then the FS, which use a different infiltration scheme. This is not a really 'coupled model' , as compared to others existing (same references as before, i.e., Burton and Bathurst, 1998, Claessens et al. 2007, Simoni et al., 2008, Arnone et al., 2011, Lepore et al., 2013, Tao and Barros, 2014). The 'initial condition' (P5L9) are updated at each time? These means that, in TRIGS, each time step is 'independent' from the previous. Honestly, I don't see the efficiency of such a system when many other models exists that do the same in a continuous way. Actually, I would directly substitute the TRIGRS infiltration model within CREST an then implement the FS equation as a function of the moisture and water table. (they are both written in fortran language); it is more complex but more efficient and functional. 7. Are all the input data listed in fig.1 fed to both the models?

RESOLUTION AND LANDSLIDE SIZE 8. Working spatial resolution is 90m (P7L18). How this size is compared to the landslide average size? Since you build up a confusion matrix, you need to convert the inventory map into raster structure. Some studies (e.g. Claessens et al., 2005 and Tarolli and Tarboton, 2006) have demonstrated the model performances also depend on the comparison of the landslide size and resolution cell (the impact on model results is mainly caused by the effects of landform parameters, i.e. slope, aspect, curvature). I believe that a description of landslide types and characteristics (size, depths) should be given in Section 2.2.1. Also, It should be reminded somewhere that such models fit well for shallow landslide.

I will lay out my SPECIFICS CONCERNS below, referring to page and line numbers.

1. P6L30. This sentence is misleading. It's true that the 'minimum forcing data' are precipitation and evapotraspiration, but it requires various parameters and input maps. Note that, other models, needs only the precipitation forcing.

2. P7L34. It is not clear to me whether the soil properties are spatialized in CREST. How the soil texture classes are involved in the model?

3. P8L14. USDA textural soil classification does not provide the mechanical properties (i.e. friction angle and cohesion). How did you estimated these? FS is obviously extremely sensitive to these parameters . . .

4. P9L27 – Are you able to explain the shift between modeled and observed peak discharges?

5. P9L29 – Which is the depth of failure? Please specify. This is crucial.

6. P9L30 – In both Fig.5a and 5b, the second basin from the left (Tuckasegee River basin?) is the one in which landslide are overpredicted . . . I guess because it's the

basin with most the steepest areas. . . Moreover, I would like to see the corresponding pore-pressure map (TRIGRS and iCRESTRIGRS)

7. P9L35 – radius 500 m means ∼78.5 ha that means ∼96 computation grid cells. This means that 'your model success' when 1 over 96 cells success . . . Again, in such analyses the landslide size should be taken into account.

8. P10L15 – Please, report the corresponding AUC values in the figure.

9. P12L2 – you could also work on FS equation and mechanical properties (a better soil characterization?). Also, as said before, model performances are very sensitive to the landslide size and characteristics, which here are not taken into account in no way.

MINOR P7L30 – it should be Fig.2a. P7L33 – please specify Fig.2c (map of soil) and Fig.2d (land cover map)

REFERENCES THAT ARE NOT IN THE MANUSCRIPT

Arnone E., Noto L. V., Lepore C., Bras R. L. 2011. Physically-based and distributed approach to analyze rainfall-triggered landslides at watershed scale, Geomorphology 133: 121-131

Burton, A., Bathurst, T.J., 1998. Physically based modeling of shallow landslide sediment yield at the catchment scale. Environmental Geology 35, 89–99.

Claessens L., Schoorl J. M., Veldkamp A. 2007. Modelling the location of shallow landslides and their effects on landscape dynamics in large watersheds: An application for Northern New Zealand, Most 87: 16 - 27

Claessens L., Heuvelink G. B. M., Schoorl J. M., Veldkamp A. 2005. DEM resolution effects on shallow landslide hazard and soil redistribution modelling, Earth Surface Processes and Landforms 30: 461-477.10.1002/esp.1155.

Lepore, C., Arnone, E., Noto, L. V., Sivandran, G. & Bras, R. L. 2013. Physically based modeling of rainfall-triggered landslides: a case study in the Luquillo forest, Puerto

Rico. Hydrol. Earth Syst. Sci. 17, 3371–3387.

Tarolli P., Tarboton D. G. 2006. A new method for determination of most likely landslide initiation points and the evaluation of digital terrain model scale in terrain stability mapping, Hydrol. Earth Syst. Sci. 10: 663-677.10.5194/hess-10-663-2006.

---

## Author Comment (AC1) · 10 Jul 2016

1. *"This study General Comments: This paper represents a significant conceptual advancement in the modeling of flash flooding and shallow, rainfall-induced landslides. Combining CREST and TRIGRS appears to provide a fairly complete solution for modeling the surface water, shallow subsurface hydrology related to runoff and flooding as well as the initial hydrologic conditions, transient pressure head, and slope stability processes related to landslide initiation. The paper is generally well written and provides a general framework for modeling the hazards resulting from heavy or prolonged rainfall. The paper could be improved by stating more of the modeling assumptions (see specific comments) used in the case study and making a number of technical corrections."*

Response: We thank the reviewer for his/her positive comments. We totally agree with the reviewer's assessment that our paper presents a significant conceptual advancement in the modeling of flash flooding and shallow, rainfall-induced landslides. More importantly, revisions based on the comments from this reviewer and the other will further improve this manuscript.

2. *"P. 1, lines 19-20 and p. 2, line 9: The phrase "leading to losses that are significantly greater than the sum of the losses from the individual hazards" seems a little nonsensical. I understand that the authors are trying to convey the synergistic effects of flooding and landslides, but the effect of combined versus separate action of the hazards needs to be expressed more clearly. Consider 'losses resulting from the combined hazards are significantly greater than the sum of losses from the hazards if acting separately'"*
Response:
We agree with this suggestion and will make corresponding change in the revised manuscript.

3. *"P.1 line 23. It is confusing to refer to this combined modeling system as an "early warning system." A modeling system is part of many early warning systems, but warning systems consist of much more than computer models. Change 'coupled flash flood and landslide disaster early warning system' to 'coupled flash flood and landslide initiation modeling system'"*
Response:
This is a valid point. We agree that it's more accurate to denote the system as a "modeling system" rather than a "warming system".

4. *"P. 2, lines 27 – 32. These two sentences seem somewhat contradictory. The emotional effects of the recent devastating tornado seems to have been more critical in determining public behavior than the source of the warnings. Either rephrase to clarify that different sources of the flash flood and storm forecasts had a lesser role or explain how this contributed to the fact that the public's attention was drawn to the*

*tornado warnings. "*
Response:
We can clarify this statement in the revised manuscript.

5.  *"P. 4, line 12. Is there a reference for the "multi-linear reservoir" concept? "*
Response:
Yes, there are one or more references for the concept. Wang et al. 2011, HSJ is one of them.

6.  *"P. 4, lines 30 – 31. Delete reference to Iverson (2000) in line 31. The 'pressure diffusion solutions for pressure changes below the water table,' though somewhat similar to the pressure-diffusion solution presented by Iverson (2000) for specified flux at the ground surface use different boundary conditions."*
Response:
We will make the correction in the revised manuscript.

7.  *"P. 6, line 6. Are the parameters a and b in equation 3 determined theoretically or empirically? "*
Response:
Parameter a can be determined using a priori estimation that is a semi-physical semi-empirical method. Details can be found in Yao et al. 2012 (Journal of Hydrology, http://dx.doi.org/10.1016/j.jhydrol.2012.08.025). Parameter b is a conceptual parameter and is determined empirically, but it is a relatively insensitive parameter.

8.  *"P. 8, lines 10 – 20. A few more details are needed regarding modeling assumptions: What assumptions were made regarding soil depth (constant or spatially varying? If so, how?)? Did you check the factor of safety for prestorm initial conditions to confirm that no (or very few) grid cells had a factor of safety less than 1? For*

*the pressure head computations used to compute factor of safety, did TRIGRS and iCRESTRIGRS use the unsaturated infiltration model as described at the bottom of p. 4? If so, why does table 1 not contain columns for the alpha (inverse height of capillary fringe) and residual moisture content values used in computing pressure head?"*
Response:

Regarding the detailed information the reviewer asked, we will add them into the revised manuscript. The soil depth data used in this study were derived from the SSURGO data and vary spatially. We confirm that no grid cells had a factor of safety less than 1 prior the storm in our simulations. The unsaturated infiltration model was used in TRIGRS and CRESTRIGRS. Because the alpha parameter is a fitting parameter and we don't have its spatial information, we, therefore, treated this parameter a lumped parameter in this study.

9. *"P. 12, lines 6 – 8. A new MPI version of TRIGRS is available that could help with the large area and finer grid assessments suggested here. The citation is Alvioli, M., and Baum, R.L., 2016, Parallelization of the TRIGRS model for rainfall-induced landslides using the message passing interface: Environmental Modelling Software, Vol. 81, July, p. 122 - 135. doi:10.1016/j.envsoft.2016.04.002 "*
Response:

Thanks for providing the reference. We will add it into the discussion section.

10. *"Figure 6. Please label key factor of safety values (0.9, 1.0, 1.1) along the ROC curve."*
Response:

We accept this suggestion and will add the labels in the revised manuscript.

11. *"Technical Corrections: P. 2, lines 16 – 17. Change 'damaging infrastructure based on the work of Wooten et al. (2008) and the following geological surveys." to "damaging infrastructure (Wooten et al. 2008; Bauer et al. 2012).' P. 2, line 27.*

*Change 'marking' to 'making' P. 2, line 28. Change 'largely' to 'partly' P. 2, line 29. Change '(Uccellini et al., 2014); the public's attention' to '(Uccellini et al., 2014). Moreover, the public's attention' P. 6, line 21. Change 'model realistically compute' to 'model to realistically compute' P. 7, line 34. Insert 'dataset' after '(STATSGO)' P. 8, line 2. Change 'Land Cover Database (NLDC) 2011 land cover database (Homer et al., 2015).' to 'Land Cover Database (NLDC) 2011 (Homer et al., 2015).' P. 8, lines 4 – 6. Change '15-minute streamflow observations from four USGS streamflow gauges (03503000 at Little Tennessee River, 03513000 at Tuckasegee River, 03460795 at Pigeon River, and 03453500 at French Broad River) were aggregated to hourly resolution and serve as streamflow validation data for the model.' to 'Streamflow observations from four USGS streamflow gauges (03503000 at Little Tennessee River, 03513000 at Tuckasegee River, 03460795 at Pigeon River, and 03453500 at French Broad River) were aggregated from 15-minute to hourly resolution and serve as streamflow validation data for the model.' P. 9, line 11. Insert 'indicator' after 'global statistical accuracy' P. 9, lines 19 – 20. Change "Apparently, the storm was rapid and intense." to "The storm was rapid and intense." P. 9, line 29. Change "the land surface's slope is predicted to fail" to "the regolith that covers the sloping ground surface is predicted to fail" P. 10, line 7. Change "way how infiltration" to "way that infiltration" P. 11, line 34. Change "It is worth to note that there is still a large room for improving" to "It is worth noting that there is still much room for improving" P. 13, line 1. Change "and Anders, C. F.:" to "and Anderson, G. F.:""*

Response:

We appreciate that the reviewer carefully read through our manuscript and pointed out these valuable details. We agree that most of these technical corrections are appropriate and will improve the quality of this manuscript. Therefore, we will revise these sentences and/or statements in the revised manuscript.

---

## Author Comment (AC2) · 10 Jul 2016

1. *"I read with interest this research paper which describes a coupled system for prediction of landslide occurrences triggered by intense precipitation. The system is based on the use, in a cascade manner, of the models CREST and TRIGRS. Finally, the authors describe the results of an interesting case study. In the Introduction, the issues of flood and landslide disasters are well described in the context of natural hazards, which characterize the study area. The general objects of the work are clear and well stated, the proposed methodology is of good scientific interest, and also the presentation quality is satisfactory. However, I have some doubts on the efficiency of the created system and I argue the use of the terms 'early warning system' and*

*'flood'."*

Response:

We thank the reviewer for his/her positive and constructive comments. We agree with the reviewer that our study is of good scientific interest and have a good quality. In terms of the naming of this system, we agree that the system we developed is a modeling system rather than an early warning system. However, we only called the system as an "early warning system" once in the manuscript. We will change the statement and correct the wording in the revised manuscript. Regarding the word "flood" used in this study, please see Item 2 of Responses to Reviewer 2 (i.e. the following item).

2.     *"I am not sure if using the term 'flood' is here appropriate. If I understood correctly, CREST model predict the discharge at outlet and the spatially distributed surface runoff. I did not see here any flood disaster forecasting analysis (as claimed by the authors), which normally expects identification of floodplains, critical discharges, etc.. Does the model include a propagation module? Also, note that in flash flood events, the evapotraspiration is almost negligible, whereas the spatially distributed component of CREST model mainly derives, if I understood correctly, from the use of LAI and vegetation cover, from which evapotraspiration and interception processes are estimated."*

Response:

We respectively insist that the term 'flood' used in our modeling system is appropriate. It is widely known in our field that hydrological models, either lumped models or distributed models, are used to simulate streamflow (discharges) at the outlet or any grid cell. These simulated streamflow values can be used to detect the risks and occurrences of floods based on flood frequency analysis of retrospective model simulations. Based on this idea, several well-known global or regional flood forecasting/monitoring systems have been established (please refer to http://www.gdacs.org/flooddetection/, http://flash.ou.edu, http://flood.umd.edu, and http://eos.ou.edu). In addition, the

CREST model is able to simulate discharges at any stream cell and surface flow at any grid cell. We didn't represent much flood analysis simply because we didn't run the model long enough to generate enough samples for this type of analysis. The CREST model doesn't have a propagation module but it does have a routing module. In terms of evapotranspiration, it is trivial for the simulation of a flash flood event. However, it impacts the long-term water balance, which can impact the accurate simulation of hydrograph in turn.

3. *"Similarly, an early warning system is a more complex system which includes the definition of a 'chain of different communication systems working together and aimed at the detection, analysis and mitigation of potentially hazardous events'. For example, they normally includes a monitoring network for real-time analysis and the definition of thresholds for launching alert/waning signals. I would simply say that the developed system could be potentially used within an early warning system' (P1L23)"*
Response:
We agree that "an early warming system" is more complicated than a modeling system. Like we mentioned in Item 1 of Responses to Reviewer 2, we only called the system as an "early warning system" once in the manuscript. We will change the statement and correct the wording in the revised manuscript to make it more accurate.

4. *"Authors state that 'studies on dynamically coupling hydrological processes predicted by distributed hydrological models with soil physics and mechanics determining slope stability are still in a very early stage' (see discussion at P3L16-25). Actually I do not agree with authors analysis. Literature provides many examples of spatial distributed and coupled hydrological-stability models, which have been successfully utilized. Besides the ones cited by the authors (Simoni et al., 2008 and Lanni et al., 2012), authors can refer, for example, to Burton and Bathurts (1998), Claessens et al. (2007), Arnone et al. (2011), Lepore et al. (2013), Tao and Barros (2014) (these last two are also cited and described by Bogard and Greco (2014) as 'good examples' of coupled physically based hydrological and slope stability modeling at catchment*

*scale). Also, you may more deeply discuss the sentence 'due to lack of knowledge of interactions between these processes and differences in the spatiotemporal scales of the flood and landslide events'."*

Response:

We mean that there is still large room for studies on the coupling of distributed hydrological model with the landslide model. We will relax corresponding statements and discussions, and cite additional appropriate references in the revised manuscript.

5. *"If I understood correctly, CREST model does use only precipitation, LAI and vegetation cover as distributed in input variables (as said above). Other model parameters I guess are homogenous across the watershed. How about the soil parameters that influence the infiltration processes, and slope stability, such as porosity, hydraulic conductivity, retention curve parameters? Please specify whether these are distributed or homogeneous."*

Response:

Most of model parameters for the CREST model in this study are distributed, including the soil parameters that influence the infiltration processes, and slope stability. These parameter values at each grid cell were determined by its soil class through a look-up table (Table 1). We described how we determined these parameters in Section 2.2.3. We can provide further details on how these distributed parameter values are determined if necessary.

6. *"There is no mention on how FS is computing and which are the parameters affecting stability. I would like to see the FS equation, explain whether is based on classical Mohr-Coulomb or Bishop failure criterion (for unsaturated conditions) and if it is computed ad a fixed soil thickness of different soil depths. Indeed, FS varies significantly with depths, at given parameters (see other comments below)."*

Response:

We followed the way that TRIGRS computes FS and the depth of slope initiation. FS

is calculated as the ratio of resisting basal Coulomb friction to gravitationally induced down-slope basal driving stress (Baum et al. 2010, JGR, doi:10.1029/2009JF001321). FS is calculated for transient pressure heads at multiple depths, Z . Failure is predicted when FS < 1 and stability holds where FS >= 1. Thus, the depth Z where FS first drops below 1 will be the depth of landslide initiation (see Baum et al. 2010, JGR, doi:10.1029/2009JF001321).

In the unsaturated zone, a simple approximation for Bishop's (1959) effective stress parameter ($\chi$) is used in computing the factor of safety. To compute the factor of safety above the water table, the matric suction in the FS computation equation is multiplied by $\chi$. More details can be found in Baum et al. 2010, JGR.

7.    *"The integration between the two models is not very clear to me. TRIGRS model has its own infiltration module which is based on an analytical solution of the Richards equation for vertical infiltration. As input, it requires rainfall intensity; based on the hydraulic properties, which vary in space, it computes the infiltration rate and thus the pore-pressure (or soil moisture), which are used to compute the FS. My question is, how this framework interfaces with CREST model? My impression is that the two models are simply used in 'cascade' and therefore they are run separately, to evaluate first the runoff, and then the FS, which use a different infiltration scheme. This is not a really 'coupled model' , as compared to others existing (same references as before, i.e., Burton and Bathurst, 1998, Claessens et al. 2007, Simoni et al., 2008, Arnone et al., 2011, Lepore et al., 2013, Tao and Barros, 2014). The 'initial condition' (P5L9) are updated at each time? These means that, in TRIGS, each time step is 'independent' from the previous. Honestly, I don't see the efficiency of such a system when many other models exists that do the same in a continuous way. Actually, I would directly substitute the TRIGRS infiltration model within CREST an then implement the FS equation as a function of the moisture and water table. (they are both written in fortran language); it is more complex but more efficient and functional."*
Response:

We coupled the CREST model with TRIGRS model through a one-way coupling approach. The CREST model provides all hydrologic storages and fluxes, including interception by vegetation, infiltration, runoff generation, water routing, and re-infiltration of excess surface runoff from upstream cells to downstream cells, and provides the initial conditions, e.g. soil wetness and depth of water table. Therefore, the CREST model provides the infiltration for the TRIGRS model. The initial condition of the TRIGRS model is not necessarily updated at each time step. Instead, the coupled model can automatically determine the beginning of each storm. During the course of the storm, all of time steps share the same initial condition.

8. *"Are all the input data listed in fig.1 fed to both the models?"*
Response:
No. We will revise the figure to show the accurate data flow in the revised manuscript.

9.   *"Working spatial resolution is 90m (P7L18). How this size is compared to the landslide average size? Since you build up a confusion matrix, you need to convert the inventory map into raster structure. Some studies (e.g. Claessens et al., 2005 and Tarolli and Tarboton, 2006) have demonstrated the model performances also depend on the comparison of the landslide size and resolution cell (the impact on model results is mainly caused by the effects of landform parameters, i.e. slope, aspect, curvature). I believe that a description of landslide types and characteristics (size, depths) should be given in Section 2.2.1. Also, It should be reminded somewhere that such models fit well for shallow landslide."*
Response:
The reviewer is right that model performance also depends on the landslide size and the resolution of grid cells. We don't have the size information for all landslides. When we compared the model simulations with the observations, we think that the model correctly captures the landslide event if one or more 90-m grid cells within a radius of 250 m ( 3 grid cells) around the center of the recorded landslide have FS<1. The radius is roughly based on the average landslide size. We can provide more information on

the landslide sizes and types.

SPECIFICS CONCERNS:

10. *"P6L30. This sentence is misleading. It's true that the 'minimum forcing data' are precipitation and evapotraspiration, but it requires various parameters and input maps. Note that, other models, needs only the precipitation forcing."*
Response:
We respectively argue that the parameters and input maps are static input data not the forcing data. The "minimum forcing data" for this model indeed just include the precipitation and evapotranspiration. In addition, we don't comment the other models here.

11. *"P7L34. It is not clear to me whether the soil properties are spatialized in CREST. How the soil texture classes are involved in the model?"*
Response:
As we mentioned in Item 5 of Responses to Reviewer 2, the soil properties are distributed and derived from the SSURGO dataset (please refer to Section 2.2.3 and Fig. 2c). We built a look-up table to assign specific parameter values to each soil class (Table 1) based on valued provide in the literature. We then used the soil class of each grid cell to determine the parameter values of the same grid cell.

12. *"P8L14. USDA textural soil classification does not provide the mechanical properties (i.e. friction angle and cohesion). How did you estimated these? FS is obviously extremely sensitive to these parameters?"*
Response:
The friction angle and cohesion values of the USDA textural soil classes were derived as the means of the ranges determined from Das (2008), Hough (1969), Terzaghi et al. (1996) and (Dysli, 2000) (please refer to Table 1).

13. *"P9L27 – Are you able to explain the shift between modeled and observed peak*

*discharges?"*

Response:

There are a couple of possible reasons causing this shift. It could be due to the uncertainty in the routing scheme of this model. It could be caused by the uncertainty in the spatial distribution of rainfall data. We can add discussion on the possible causes in the revised manuscript.

14. *"P9L29 – Which is the depth of failure? Please specify. This is crucial."*

Response:

Depth of failure calculated by the CRESTRIGRS and TRIGRS models vary spatially and temporally.

15. *"P9L30 – In both Fig.5a and 5b, the second basin from the left (Tuckasegee River basin?) is the one in which landslide are overpredicted. I guess because it's the basin with most the steepest areas. Moreover, I would like to see the corresponding pore-pressure map (TRIGRS and iCRESTRIGRS)"*

Response:

We can provide the additional information in the revised manuscript.

16. *"P9L35 – radius 500 m means 78.5 ha that means 96 computation grid cells. This means that 'your model success' when 1 over 96 cells success. Again, in such analyses the landslide size should be taken into account."*

Response:

There is one mistake here. We mean a radius of 250 m (a perimeter of 500m). We will provide more information on the landslide size and further consider its impact in the revised manuscript.

17. *"P10L15 – Please, report the corresponding AUC values in the figure."*
Response:
We can add the AUC values in the figure in the revised manuscript.

18. *"P12L2 – you could also work on FS equation and mechanical properties (a better soil characterization?). Also, as said before, model performances are very sensitive to the landslide size and characteristics, which here are not taken into account in no way."*
Response:
We agree that these are good points to be mentioned in the discussion section.

19. *"MINOR P7L30 – it should be Fig.2a. P7L33 – please specify Fig.2c (map of soil) and Fig.2d (land cover map)"*
Response:
Thanks for pointing this out. We will correct it and specify Figs. 2c,d in the revised manuscript.

---

## Author Response (AR1)

**Response to Editor's Remarks:**

*"Thanks to the reviewers and you for excellent discussion on your paper. It became clear your work is evaluated as a significant contribution to hydrology and landslide research. It also fits perfectly in HESS. However, there have been several technical issues -mainly extra explaining and terminology- to be addressed, most of which you already replied on and agreed to add to the paper. I want to stress that it is important to be as open, clear and complete as possible in terminology, pro- and cons of approach and possible application, explicitly discuss the way the model integration is done, etc. Both reviewers mentioned these points. This will help readers to use it, work with it and develop it."*

Response:
We thank the editor for his positive comments and assessments. We have substantially revised our manuscript based on your remarks and the reviewer comments. In particular, we resolved the technical issues by adding additional explanation and clarifying terminology and the model integration. For more information on the revision, please refer to the following point-to-point reply to the review.

**Response to Anonymous Referee #1:**

1. *"This study General Comments: This paper represents a significant conceptual advancement in the modeling of flash flooding and shallow, rainfall-induced landslides. Combining CREST and TRIGRS appears to provide a fairly complete solution for modeling the surface water, shallow subsurface hydrology related to runoff and flooding as well as the initial hydrologic conditions, transient pressure head, and slope stability processes related to landslide initiation. The paper is generally well written and provides a general framework for modeling the hazards resulting from heavy or prolonged rainfall. The paper could be improved by stating more of the modeling assumptions (see specific comments) used in the case study and making a number of technical corrections."*

Response:
We thank the reviewer for his/her positive comments. We totally agree with the reviewer's assessment that our paper presents a significant conceptual advancement in the modeling of flash flooding and shallow, rainfall-induced landslides. More importantly, revision based on the comments from this reviewer and the other has further improved this manuscript.

2. *"P. 1, lines 19-20 and p. 2, line 9: The phrase "leading to losses that are significantly greater than the sum of the losses from the individual hazards" seems a little nonsensical. I understand that the authors are trying to convey the synergistic effects of flooding and landslides, but the effect of combined versus separate action of the hazards needs to be expressed more clearly. Consider 'losses resulting from the combined hazards are significantly greater than the sum of losses from the hazards if acting separately'"*

Response:
We accepted this suggestion and made the corresponding change in the revised manuscript (see page 1, lines 18-20 and page 2, lines 5-7).

3. *"P.1 line 23. It is confusing to refer to this combined modeling system as an "early warning system." A modeling system is part of many early warning systems, but warning systems consist of much more than computer models. Change 'coupled flash flood and landslide disaster early warning system' to 'coupled flash flood and landslide initiation modeling system'"*

Response:
We agree that it's more accurate to denote the system as a "modeling system" rather than a "warming system". We have made the corresponding changes in the revised manuscript (see page 1, line 23 and page 3, line 30).

4. *"P. 2, lines 27 – 32. These two sentences seem somewhat contradictory. The emotional effects of the recent devastating tornado seems to have been more critical in determining public behavior than the source of the warnings. Either rephrase to clarify that different sources of the flash flood and storm forecasts had a lesser role or explain how this contributed to the fact that the public's attention was drawn to the tornado warnings."*

Response:

We clarified these statements in the revised manuscript and changed them to "*This is partly due to the fact that the storm (accompanied by heavy precipitation and the tornado) and flash flood were forecasted by two separate warning systems and their warnings were issued separately (Uccellini et al., 2014); the public was well aware of the tornado threat but largely unaware of the flood threat in spite of several NWS products and outreach efforts (Uccellini et al., 2014). Moreover, the public's attention was mostly drawn to the tornado warnings (not to the flash flooding threat) mainly because this storm occurred only ten days after the disastrous EF-5 tornado which devastated Moore, OK and resulted in 24 fatalities and $2 billion in property damage*" (see page 2, lines 25-31).

5. "*P. 4, line 12. Is there a reference for the "multi-linear reservoir" concept?*"

Response:
We added the reference, i.e. Wang et al. 2011, in the revised manuscript (see page 4, line 12).

6. "*P. 4, lines 30 – 31. Delete reference to Iverson (2000) in line 31. The 'pressure diffusion solutions for pressure changes below the water table,' though somewhat similar to the pressure-diffusion solution presented by Iverson (2000) for specified flux at the ground surface use different boundary conditions.*"

Response:
We deleted the reference to Iverson (2000).

7. "*P. 6, line 6. Are the parameters a and b in equation 3 determined theoretically or empirically?*"

Response:
Symbol a in Equation 3 is not a parameter but a variable. It denotes the fraction of a grid cell whose water capacity is less than or equal to the mean water capacity of this grid cell. Parameter b is a conceptual parameter and is determined empirically, but it is a relatively insensitive parameter. We added additional explanation on the two symbols in the text (see page 6, lines 15-18).

8. "*P. 8, lines 10 – 20. A few more details are needed regarding modeling assumptions: What assumptions were made regarding soil depth (constant or spatially varying? If so, how?)? Did you check the factor of safety for prestorm initial conditions to confirm that no (or very few) grid cells had a factor of safety less than 1? For the pressure head computations used to compute factor of safety, did TRIGRS and iCRESTRIGRS use the unsaturated infiltration model as described at the bottom of p. 4? If so, why does table 1 not contain columns for the alpha (inverse height of capillary fringe) and residual moisture content values used in computing pressure head?*"

Response:
The soil depth data used in this study were derived from the SSURGO data and vary spatially. We confirm that no grid cells had a factor of safety less than 1 prior the storm

in our simulations. The unsaturated infiltration model was used in TRIGRS and CRESTRIGRS. Because the alpha parameter is a fitting parameter and we don't have its spatial information, we, therefore, treated this parameter a lumped parameter in this study.

9. "*P. 12, lines 6 – 8. A new MPI version of TRIGRS is available that could help with the large area and finer grid assessments suggested here. The citation is Alvioli, M., and Baum, R.L., 2016, Parallelization of the TRIGRS model for rainfall-induced landslides using the message passing interface: Environmental Modelling & Software, Vol. 81, July, p. 122 - 135. doi:10.1016/j.envsoft.2016.04.002* "

Response:
Thanks for providing the reference. We added the discussion and this literature in page 12, lines 30-32.

10. "*Figure 6. Please label key factor of safety values (0.9, 1.0, 1.1) along the ROC curve.*"

Response:
We accepted this suggestion and added the labels into the new Figure 6.

11. "*Technical Corrections:P. 2, lines 16 – 17. Change 'damaging infrastructure based on the work of Wooten et al. (2008) and the following geological surveys." to "damaging infrastructure (Wooten et al. 2008; Bauer et al. 2012).'*"

Response:
We made the corresponding changes per the reviewer's suggestion (see page 2, line 13).

12. "*P. 2, line 27. Change 'marking' to 'making'*"

Response:
We corrected it (page 2, line 24).

13. *P. 2, line 28. Change 'largely' to 'partly'*

Response:
We accepted the suggestion and made the corresponding change (page 2, line 25).

14. *P. 2, line 29. Change '(Uccellini et al., 2014); the public's attention' to '(Uccellini et al., 2014). Moreover, the public's attention'*

Response:
We accepted the suggestion and made the corresponding change (page 2, lines 26-27).

15. *P. 6, line 21. Change 'model realistically compute' to 'model to realistically compute'*

Response:
We accepted the suggestion and made the corresponding change (page 6, line 30).

16. *P. 7, line 34. Insert 'dataset' after '(STATSGO)'*

Response:
We accepted the suggestion and made the corresponding change (page 8, line 6).

*17. P. 8, line 2. Change 'Land Cover Database (NLDC) 2011 land cover database (Homer et al., 2015).' to 'Land Cover Database (NLDC) 2011 (Homer et al., 2015).'*

Response:
We accepted the suggestion and made the corresponding change (page 8, line 9).

*18. P. 8, lines 4 – 6. Change '15-minute streamflow observations from four USGS streamflow gauges (#03503000 at Little Tennessee River, # 03513000 at Tuckasegee River, # 03460795 at Pigeon River, and # 03453500 at French Broad River) were aggregated to hourly resolution and serve as streamflow validation data for the model.' to 'Streamflow observations from four USGS streamflow gauges (#03503000 at Little Tennessee River, # 03513000 at Tuckasegee River, # 03460795 at Pigeon River, and # 03453500 at French Broad River) were aggregated from 15-minute to hourly resolution and serve as streamflow validation data for the model.'*

Response:
We accepted the suggestion and made the corresponding change (page 8, lines 10-12).

*19. P. 9, line 11. Insert 'indicator' after 'global statistical accuracy'*

Response:
We accepted the suggestion and made the corresponding change (page 9, line 26).

*20. P. 9, lines 19 – 20. Change "Apparently, the storm was rapid and intense." to "The storm was rapid and intense."*

Response:
We accepted the suggestion and made the corresponding change (page 9, line 34).

*21. P. 9, line 29. Change "the land surface's slope is predicted to fail" to "the regolith that covers the sloping ground surface is predicted to fail"*

Response:
We accepted the suggestion and made the corresponding change (page 10, line 14).

*22. P. 10, line 7. Change "way how infiltration" to "way that infiltration"*

Response:
We accepted the suggestion and made the corresponding change (page 10, line 28).

*23. P. 11, line 34. Change "It is worth to note that there is still a large room for improving" to "It is worth noting that there is still much room for improving"*

Response:
We accepted the suggestion and made the corresponding change (page 12, line 19).

*24. P. 13, line 1. Change "and Anders, C. F.:" to "and Anderson, G. F.:""*

Response:
We have corrected it (page 13, line 41).

**Response to Anonymous Referee #2:**

1. "*I read with interest this research paper which describes a coupled system for prediction of landslide occurrences triggered by intense precipitation. The system is based on the use, in a cascade manner, of the models CREST and TRIGRS. Finally, the authors describe the results of an interesting case study. In the Introduction, the issues of flood and landslide disasters are well described in the context of natural hazards, which characterize the study area. The general objects of the work are clear and well stated, the proposed methodology is of good scientific interest, and also the presentation quality is satisfactory. However, I have some doubts on the efficiency of the created system and I argue the use of the terms 'early warning system' and 'flood'.*"

Response:
We thank the reviewer for his/her positive and constructive comments. We agree with the reviewer that our study is of good scientific interest and have a good quality. In terms of the naming of this system, we agree that the system we developed is a modeling system rather than an early warning system; thus, we have revised the corresponding statements and deleted the "early warning system" term (see page 1, line 23 and page 3, line 30). Regarding the "flood" term used in this study, please see Item 2 of Responses to Anonymous Referee #2 (i.e. the following item).

2. "*I am not sure if using the term 'flood' is here appropriate. If I understood correctly, CREST model predict the discharge at outlet and the spatially distributed surface runoff. I did not see here any flood disaster forecasting analysis (as claimed by the authors), which normally expects identification of floodplains, critical discharges, etc.. Does the model include a propagation module? Also, note that in flash flood events, the evapotraspiration is almost negligible, whereas the spatially distributed component of CREST model mainly derives, if I understood correctly, from the use of LAI and vegetation cover, from which evapotraspiration and interception processes are estimated.*"

Response:
We respectively insist that the 'flood' term used in our modeling system is appropriate. It is widely known in our field that hydrological models, either lumped models or distributed models, are used to simulate streamflow (discharges) at the outlet or any grid cell. These simulated streamflow values can be used to detect the risks and occurrences of floods based on flood frequency analysis of retrospective model simulations. Based on this idea, several well-known global or regional flood forecasting/monitoring systems have been established (please refer to http://www.gdacs.org/flooddetection/, http://flash.ou.edu, http://flood.umd.edu, and http://eos.ou.edu). In addition, the CREST model is able to simulate discharges at any stream cell and surface flow at any grid cell. We didn't represent much flood analysis simply because we didn't run the model long enough to generate enough samples for this type of analysis. The CREST model doesn't have a water-wave propagation module but it does have a routing module. The CREST/iCRESTRIGRS model is able to use LAI to estimate evapotranspiration (ET),

but we directly used the remote-sensing based ET data as input in this study (see page 7, lines 29-31). In terms of evapotranspiration, it is trivial for the simulation of a flash flood event. However, it impacts the long-term water balance, which can impact the accurate simulation of hydrograph in turn.

3. "*Similarly, an early warning system is a more complex system which includes the definition of a 'chain of different communication systems working together and aimed at the detection, analysis and mitigation of potentially hazardous events'. For example, they normally includes a monitoring network for real-time analysis and the definition of thresholds for launching alert/waning signals. I would simply say that the developed system could be potentially used within an early warning system' (P1L23)*"

Response:
We agree that "an early warming system" is more complicated than a modeling system. Like we mentioned in Item #1 of Responses to Anonymous Referee #2, we only called the system as an "early warning system" once in the manuscript. We have revised the statement and corrected the wording in the revised manuscript to make it more accurate.

4. "*Authors state that 'studies on dynamically coupling hydrological processes predicted by distributed hydrological models with soil physics and mechanics determining slope stability are still in a very early stage' (see discussion at P3L16-25). Actually I do not agree with authors analysis. Literature provides many examples of spatial distributed and coupled hydrological-stability models, which have been successfully utilized. Besides the ones cited by the authors (Simoni et al., 2008 and Lanni et al., 2012), authors can refer, for example, to Burton and Bathurts (1998), Claessens et al. (2007), Arnone et al. (2011), Lepore et al. (2013), Tao and Barros (2014) (these last two are also cited and described by Bogard and Greco (2014) as 'good examples' of coupled physically based hydrological and slope stability modeling at catchment scale). Also, you may more deeply discuss the sentence 'due to lack of knowledge of interactions between these processes and differences in the spatiotemporal scales of the flood and landslide events'.*"

Response:
We mean that there is still large room for conducting studies on the coupling of distributed hydrological model with the landslide model for large-scale flood and landslide prediction. Therefore, we revised the statement in the revised manuscript (see page3, lines 21-22). We also cited the most relevant studies listed above by the reviewer (see page 3, lines 19-21).

5. "*If I understood correctly, CREST model does use only precipitation, LAI and vegetation cover as distributed in input variables (as said above). Other model parameters I guess are homogenous across the watershed. How about the soil parameters that influence the infiltration processes, and slope stability, such as porosity, hydraulic conductivity, retention curve parameters? Please specify whether these are distributed or homogeneous.*"

Response:
Most of model parameters for the CREST model in this study are distributed, including the soil parameters that influence the infiltration processes, and slope stability. We

determined these parameter values at each grid cell according to its soil class through a look-up table (Table 1). We described how we determined these parameters in Section 2.2.3 (see page 8, lines 21-31).

6. "*There is no mention on how FS is computing and which are the parameters affecting stability. I would like to see the FS equation, explain whether is based on classical Mohr-Coulomb or Bishop failure criterion (for unsaturated conditions) and if it is computed ad a fixed soil thickness of different soil depths. Indeed, FS varies significantly with depths, at given parameters (see other comments below).*"

Response:

We followed the way that TRIGRS computes FS and the depth of slope initiation. FS is calculated as the ratio of resisting basal Coulomb friction to gravitationally induced down-slope basal driving stress (Baum et al. 2010, JGR, doi:10.1029/2009JF001321). FS is calculated for transient pressure heads at multiple depths, Z. Failure is predicted when FS < 1 and stability holds where FS >= 1. Thus, the depth Z where FS first drops below 1 will be the depth of landslide initiation (see Baum et al. 2010, JGR, doi:10.1029/2009JF001321). In the unsaturated zone, a simple approximation for Bishop's (1959) effective stress parameter ($\chi$) is used in computing the factor of safety. To compute the factor of safety above the water table, the matric suction in the FS computation equation is multiplied by $\chi$. More details can be found in Baum et al. 2010, JGR. We added the description of FS computation in the revised manuscript (see page 5, lines 11-17).

7. "*The integration between the two models is not very clear to me. TRIGRS model has its own infiltration module, which is based on an analytical solution of the Richards equation for vertical infiltration. As input, it requires rainfall intensity; based on the hydraulic properties, which vary in space, it computes the infiltration rate and thus the pore-pressure (or soil moisture), which are used to compute the FS. My question is, how this framework interfaces with CREST model? My impression is that the two models are simply used in 'cascade' and therefore they are run separately, to evaluate first the runoff, and then the FS, which use a different infiltration scheme. This is not a really 'coupled model', as compared to others existing (same references as before, i.e., Burton and Bathurst, 1998, Claessens et al. 2007, Simoni et al., 2008, Arnone et al., 2011, Lepore et al., 2013, Tao and Barros, 2014). The 'initial condition' (P5L9) are updated at each time? These means that, in TRIGS, each time step is 'independent' from the previous. Honestly, I don't see the efficiency of such a system when many other models exists that do the same in a continuous way. Actually, I would directly substitute the TRIGRS infiltration model within CREST and then implement the FS equation as a function of the moisture and water table. (they are both written in fortran language); it is more complex but more efficient and functional.*"

Response:

It is obvious that there is some misunderstanding here. We coupled the CREST model with TRIGRS model through a one-way coupling approach rather than running separately. The CREST model simulates all hydrologic storages and fluxes, and feeds the infiltration

and its history and soil moisture condition into the TRIGRS model; the TRIGRS model is then implemented to compute pore-pressure and slope stability correspondingly. The soil moisture condition at the beginning of a storm and the infiltration history during the storm is required for the TRIGRS model. In the iCRESTRIGRS model, this information is recorded automatically at each time. This updating is simple and computationally efficient. In other words, each time step is dependent on the previous time step. The detailed description on the model integration is provided in Section 2.2.3, in particular page5, lines 6-11 and page 6, lines 33-35.

8. "*Are all the input data listed in fig.1 fed to both the models?*"

Response:
No. We have revised Figure 1 to show the accurate data flow in the revised manuscript.

9. "*Working spatial resolution is 90m (P7L18). How this size is compared to the landslide average size? Since you build up a confusion matrix, you need to convert the inventory map into raster structure. Some studies (e.g. Claessens et al., 2005 and Tarolli and Tarboton, 2006) have demonstrated the model performances also depend on the comparison of the landslide size and resolution cell (the impact on model results is mainly caused by the effects of landform parameters, i.e. slope, aspect, curvature). I believe that a description of landslide types and characteristics (size, depths) should be given in Section 2.2.1. Also, It should be reminded somewhere that such models fit well for shallow landslide.*"

Response:
The reviewer is right that model performance also depends on the landslide size and the resolution of grid cells. We accepted the reviewer's suggestion to evaluate the model by accounting for the actual landslide size. Therefore, we requested more survey data, mainly the slope movement outline data, from North Carolina Geological Survey. We re-evaluated the model performance using the movement outline data. In the evaluation procedure, we regard that the model successfully predicts a real landslide if one or more cells that overlap with the slope movement outline of this landslide have a FS value of < 1; the model fails to capture a landslide event if none of these cells overlapping the outline of this landslide is modeled to be unstable (see page 10, lines 17-22). Because majority of the slope movement outlines have a long and narrow shape rather than a circular shape, which was assumed in the previous manuscript, and are distributed along the hill slopes, we achieved better values for all evaluation metrics (see page 10, lines 28-31). We provided additional information on the landslide size in the text (see page 8, lines 14-18). The other information such as landslide types and depths are unfortunately unavailable.

10. "*P6L30. This sentence is misleading. It's true that the 'minimum forcing data' are precipitation and evapotraspiration, but it requires various parameters and input maps. Note that, other models, needs only the precipitation forcing.*"

Response:
We respectively argue that the parameters and input maps are static input data not the

forcing data. The "minimum forcing data" for this model indeed just include the precipitation and evapotranspiration. In addition, we don't mean to comment the other models here. We just simply describe the minimum requirement of forcing for this model.

11. "*P7L34. It is not clear to me whether the soil properties are spatialized in CREST. How the soil texture classes are involved in the model?*"

Response:
As we mentioned in Item 5 of Responses to Anonymous Referee #2, the soil properties are distributed and derived from the SSURGO dataset (please refer to Section 2.2.3 and Fig. 2c). We built a look-up table to assign specific parameter values to each soil class (Table 1) based on valued provide in the literature. We then used the soil class of each grid cell to determine the parameter values of the same grid cell.

12. "*P8L14. USDA textural soil classification does not provide the mechanical properties (i.e. friction angle and cohesion). How did you estimated these? FS is obviously extremely sensitive to these parameters?*"

Response:
The friction angle and cohesion values of the USDA textural soil classes were derived as the means of the ranges determined from Das (2008), Hough (1969), Terzaghi et al. (1996) and (Dysli, 2000). The mechanical properties of the USDA textural soil classes are summarized in Table 1.

13. "*P9L27 – Are you able to explain the shift between modeled and observed peak discharges?*"

Response:
There are a couple of possible reasons causing this shift. It could be due to the uncertainty in the routing scheme of this model. It could be caused by the uncertainty in the spatial distribution of rainfall data. We added more discussion on the possible causes in the revised manuscript (see page 10, lines 7-10).

14. "*P9L29 – Which is the depth of failure? Please specify. This is crucial.*"

Response:
Depth of failure calculated by the CRESTRIGRS and TRIGRS models vary spatially and temporally.

15. "*P9L30 – In both Fig.5a and 5b, the second basin from the left (Tuckasegee River basin?) is the one in which landslide are overpredicted. I guess because it's the basin with most the steepest areas. Moreover, I would like to see the corresponding pore-pressure map (TRIGRS and iCRESTRIGRS)*"

Response:

The reviewer is right that landslides in the Tuckasegee River basin are overpredicted are mainly due to the steep slopes in this basin (see Fig. 2b). In terms of the pore-pressure map, we already provided these maps in Fig. 8. We think that it will be redundant to plot them again in Fig. 5.

16. "*P9L35 – radius 500 m means _78.5 ha that means _96 computation grid cells. This means that 'your model success' when 1 over 96 cells success. Again, in such analyses the landslide size should be taken into account.*"

Response:
There is one mistake here. We mean a radius of 250 m (a perimeter of 500m). Anyway, we have redone the evaluation procedure by accounting for the landslide size in the revised manuscript (please refer to Point 9 of Response to Anonymous Referee #2).

17. "*P10L15 – Please, report the corresponding AUC values in the figure.*"

Response:
We added the AUC values in this figure in the revised manuscript (see Fig. 6).

18. "*P12L2 – you could also work on FS equation and mechanical properties (a better soil characterization?). Also, as said before, model performances are very sensitive to the landslide size and characteristics, which here are not taken into account in no way.*"

Response:
We accepted the reviewer's suggestion by changing the statement on P12L2 to "*In addition, better characterization of soil properties, improved model formulation, and more information on fine-scale topography can help improve model accuracy. In particular, slope stability is highly dependent on the slope, which is highly dependent on resolution*" (page 12, lines 22-25). In addition, we re-did the model evaluation using the observed landslide size (see Point 9 of Response to Anonymous Referee #2).

19 "*MINOR P7L30 – it should be Fig.2a. P7L33 – please specify Fig.2c (map of soil) and Fig.2d (land cover map)*"

Response:
Thanks for pointing this out. We have corrected the error and cited Fig. 2c and Fig. 2d in the revised manuscript (see page 8, lines 1-12).

References:

[revised manuscript text omitted]

KZ 9/30/2016 9:09 PM

KZ 9/30/2016 9:09 PM

KZ 9/30/2016 9:09 PM